# Structural and quantum chemical analysis of exciton coupling in homo- and heteroaggregate stacks of merocyanines

David Bialas[1], André Zitzler-Kunkel[1], Eva Kirchner[1], David Schmidt[1] & Frank Würthner[1]

Exciton coupling is of fundamental importance and determines functional properties of organic dyes in (opto-)electronic and photovoltaic devices. Here we show that strong exciton coupling is not limited to the situation of equal chromophores as often assumed. Quadruple dye stacks were obtained from two bis(merocyanine) dyes with same or different chromophores, respectively, which dimerize in less-polar solvents resulting in the respective homo- and heteroaggregates. The structures of the quadruple dye stacks were assigned by NMR techniques and unambiguously confirmed by single-crystal X-ray analysis. The heteroaggregate stack formed from the bis(merocyanine) bearing two different chromophores exhibits remarkably different ultraviolet/vis absorption bands compared with those of the homoaggregate of the bis(merocyanine) comprising two identical chromophores. Quantum chemical analysis based on an extension of Kasha's exciton theory appropriately describes the absorption properties of both types of stacks revealing strong exciton coupling also between different chromophores within the heteroaggregate.

[1] Universität Würzburg, Institut für Organische Chemie and Center for Nanosystems Chemistry, Am Hubland, 97074 Würzburg, Germany. Correspondence and requests for materials should be addressed to F.W. (email: wuerthner@chemie.uni-wuerzburg.de).

The molecular exciton theory pioneered by Kasha[1] and Davydov[2] in the 1960s has contributed tremendously to the understanding of the optical properties of dye aggregates. Hence, in the last decades, a plethora of dye aggregates[3–8] have been realized and considerable efforts have been invested to rationalize the optical properties of dye self-assemblies based on exciton coupling[9–14]. However, these research activities are mainly focused on homoaggregates (that is, dye aggregates comprising the same type of chromophores), while heteroaggregates with different types of chromophores have rarely been approached[15,16]. This is actually quite surprising since self-assembly of different chromophores into heteroaggregates can lead to systems with exceptional properties that cannot be achieved by using chromophores of the same type[17,18]. Notably, it is often a common perception that exciton coupling between different types of chromophores is weak, and thus changes of the optical properties of heteroaggregates are supposed to be insignificant[19], or arise only from exciton coupling between the same type of chromophores within the heteroaggregates[20]. On the contrary, optical evidence for strong coupling between different chromophores was reported for head-to-tail connected squaraine dye co-polymers[21] as well as for mixed films composed of different dyes[22,23]. Very recently, we have also communicated that intramolecular exciton coupling between two different chromophores within a bis(merocyanine) foldamer is quite strong leading to significant changes in absorption properties of the dyes[24], which motivates for further investigations.

Reliable studies on the exciton coupling require distinct orientations and distances between the chromophores, which is difficult to realize for self-assembled supramolecular aggregates of π-conjugated dyes because the prevailing dispersion interactions between π-systems often enable various π-π-stacking arrangements of similar energy[25,26]. Therefore, more sophisticated strategies are needed to obtain discrete π-stacks like, for example, the usage of coordination cages[27], co-crystallization with appropriate spacer molecules[28] or organic molecular beam epitaxy[29]. In this regard, merocyanine dyes are promising as they enable the design of well-defined aggregate structures due to their dipolar nature that results in strong dipole–dipole interactions with high directionality[30]. Hence, this class of dyes tend to self-assembly in non-polar solvents forming dye stacks with an antiparallel orientation of the chromophores, leading to an annihilation of the overall dipole moment of the dye stack limiting the size of the aggregates[31]. In this way well-defined aggregate structures can be obtained, which can act as model systems for studies on exciton coupling between chromophores within a stack.

Keeping this in mind, we have designed bis(merocyanine) dyes 1 and 2 in which two merocyanine chromophores of same or different conjugation lengths are tethered by a rigid naphthalene spacer. Indeed, both bis(merocyanine) dyes self-assemble into highly defined dimer aggregates in less-polar solvents resulting in stacks of four chromophores, respectively. While for the symmetric bis(merocyanine) 1 only one type of dimer structure is possible, that is, homodimer, in principle three different types of heterodimers can be formed by self-assembly of the unsymmetric bis(merocyanine) 2. For the purpose of structural elucidation of the predominantly formed heteroaggregate by [1]H nuclear magnetic resonance (NMR) spectroscopy, an ethyl group was attached to the thiophene unit of the short chromophore of bis(merocyanine) 2, which self-assembles exclusively into heterodimers with the long chromophores located in the interior and the shorter ones in the terminal of the quadruple stack. The structures of the homo- and heteroaggregate dimers have been assigned by different NMR techniques and unambiguously confirmed in the solid state by X-ray analysis. The heteroaggregate dye stacks exhibit unexpected absorption spectral features compared with those of the respective monomeric dyes and the homoaggregate stacks indicating strong exciton couplings. Simulated absorption spectra of the homo- and heteroaggregate dimers obtained by time-dependent DFT calculations (TDDFT) resemble the respective experimental spectra and their spectral differences could be rationalized by extending Kasha's exciton theory to heteroaggregates revealing strong exciton coupling also between different types of chromophores within the quadruple dye stack of bis(merocyanine) 2.

## Results

**Synthesis.** The symmetric bis(merocyanine) 1 (Fig. 1a) was synthesized via Knoevenagel condensation reaction of literature known napthalenedimethylene-bridged bis(pyridone) 3 (ref. 31) and aminothiophene 4 (refs 32,33; Fig. 1c) in 55% yield. The synthesis of the unsymmetrical bis(merocyanine) 2 (Fig. 1b) bearing two different merocyanine chromophores that are tethered by the same spacer unit as in 1 could also be accomplished by condensation of bis(pyridone) 3 with the aminothiophenes 5 and 6 in a one-pot reaction. While aminothiophene 6 is literature known[34], the unknown derivative 5 was synthesized in three steps starting with the commercially available 3-ethylthiophene 7 as outlined in Fig. 1c (bottom). In the first step, Vilsmaier–Haack formylation of 7 according to a literature known procedure[35] led to the formation of a 3:1 regioisomeric mixture of 3-ethylthiophene-2-carboxaldehyde 8 and 4-ethylthiophene-2-carboxaldehyde 9. The undesired regioisomer 9 could not be separated by column chromatography due to similar polarity of both isomers. Thus, the obtained mixture was brominated to give the corresponding regioisomeric bromothiophene derivatives 10 and 11 again in a 3:1 mixture, which was then subjected to a microwave reaction with dibutylamine resulting in exclusive formation of aminothiophene 5 by selective amination of the major bromothiophene isomer 10, where the bromine substituent is sterically less encumbered. The structures of the hitherto unknown bis(merocyanine) dyes 1 and 2 were confirmed by [1]H and [13]C NMR spectroscopy, high-resolution mass spectrometry (electrospray ionization), and single-crystal X-ray analysis. For the synthetic procedure of all new compounds see the Supplementary Methods. The corresponding NMR spectra are depicted in Supplementary Figs 1–7.

**Concentration-dependent UV/vis studies.** The aggregation properties of bis(merocyanine) dyes 1 and 2 have been explored by concentration-dependent UV/vis studies in chlorobenzene. At a low concentration ($c = 4.4 \times 10^{-7}$ M), bis(merocyanine) 1 shows a typical absorption spectrum with a maximum at 541 nm for the utilized monomeric merocyanine chromophore (Fig. 2a)[34]. Upon increasing the concentration (up to $6.0 \times 10^{-4}$ M), the absorption band at 541 nm decreases with concomitant appearance of a new, hypsochromically shifted absorption band at 490 nm, which is an indication for the formation of strongly exciton coupled H-type aggregates[1]. In contrast to the symmetrical bis(merocyanine) 1, the monomer solution of unsymmetrical bis(merocyanine) 2 in chlorobenzene at a low concentration ($c = 5.0 \times 10^{-7}$ M) displays two absorption bands with maxima at 549 and 663 nm (Fig. 2b), which closely resemble those of the individual merocyanine chromophores[34]. Upon increasing the concentration (up to $3.4 \times 10^{-4}$ M), both bands decrease in an analogous manner, while a new broad band at 517 nm appears concurrently. This hypsochromically shifted band signifies the formation of excitonically coupled H-type aggregates. For the

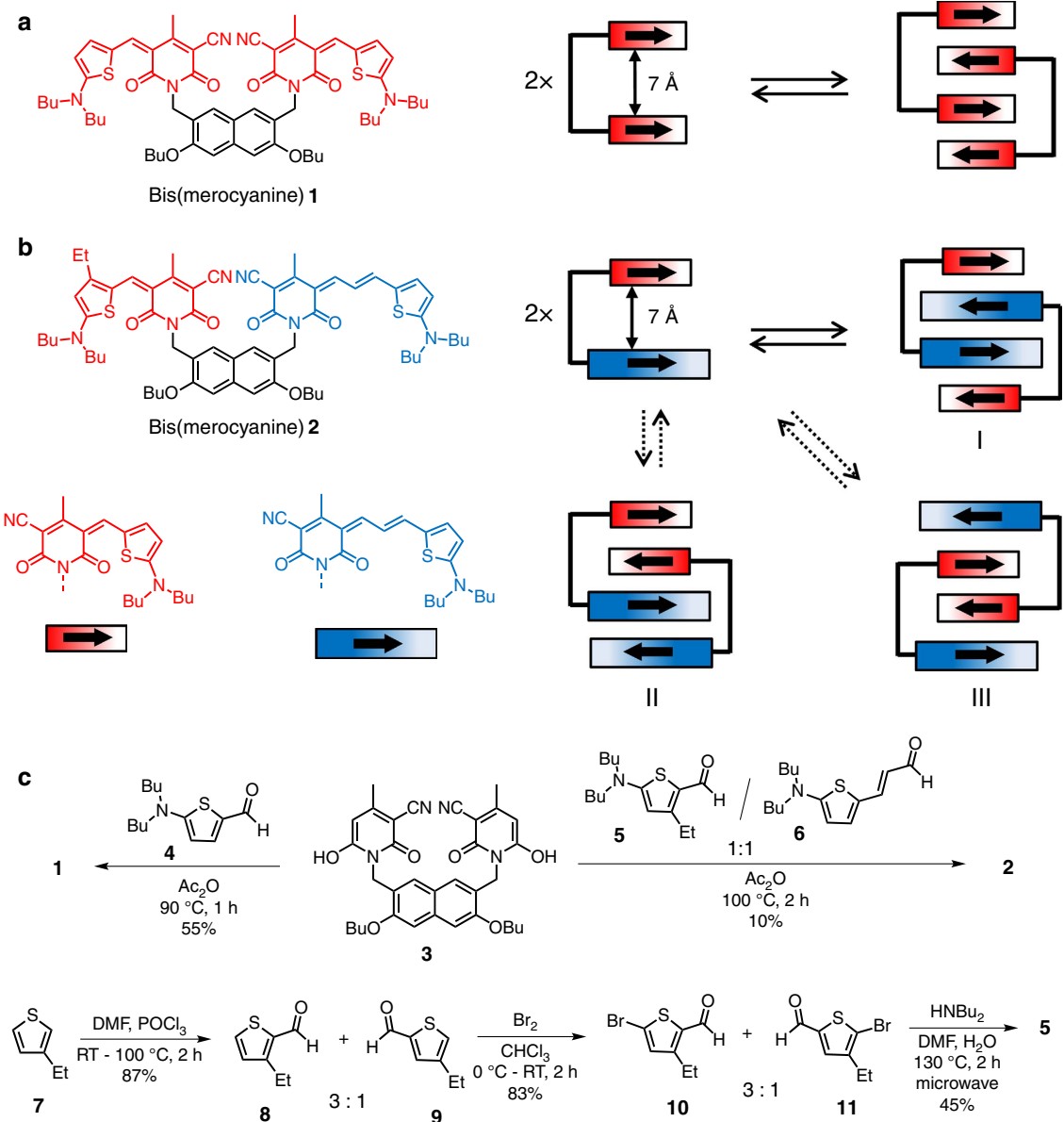

**Figure 1 | Chemical structures, synthesis and schematic representation of the self-assembly of bis(merocyanine) dyes.** (**a**) Chemical structure of bis(merocyanine) **1** with two identical chromophores and its self-aggregation into homodimer stack. (**b**) Chemical structure of bis(merocyanine) **2** with two different chromophores and its self-aggregation into three possible dimer structures. The bold arrows illustrate ground state dipole moments of the chromophore units. (**c**) Synthesis of symmetric bis(merocyanine) **1** and unsysmmetric bis(merocyanine) **2**, and the literature unknown precursor aminothiophene **5**.

concentration-dependent spectra of both bis(merocyanine) dyes clear isosbestic points at 510 and 579 nm (for **1**) and at 538, 580 and 691 nm (for **2**) were observed, revealing an equilibrium between two species, respectively, these are the monomer and the respective dimer. Indeed, the concentration-dependent UV/vis absorption data of bis(merocyanine) dyes **1** and **2** could be properly fitted to the dimer model[31] (fitting curves are shown in the inset of Fig. 2a), corroborating the formation of bimolecular complexes in both cases. Thus, a dimerization constant ($K_D$) of $5.04 \times 10^5 \, M^{-1}$ (for **1**) and $2.24 \times 10^4 \, M^{-1}$ (for **2**) in chlorobenzene and Gibbs free energies ($\Delta G_D^0$) of $-32.5$ and $-24.8 \, kJ \, mol^{-1}$, respectively, were calculated based on the dimer model. With this, the ideal monomer (turquoise line) and dimer (magenta line) spectra could be calculated (Fig. 2). The calculated dimer spectra of bis(merocyanine) **1** (Fig. 2a, magenta line)

exhibits an intense absorption band at 490 nm with a very weak transition at 573 nm, while the calculated dimer spectrum of **2** (Fig. 2b, magenta line) shows two absorption bands at 517 and 678 nm, the latter being of considerably strong intensity. While the dimer spectrum of **1** can easily be rationalized based on Kasha's exciton coupling theory (only developed for homoaggregates), the situation for **2** is obviously more complicated and does not enable us to relate the UV/vis spectra to one of the structural models I, II or III shown in Fig. 1b.

**NMR studies**. To elucidate the structural features of bis (merocyanines) **1** and **2** aggregates, in-depth one- and two-dimensional (2D) NMR studies were performed. The corresponding 2D NMR spectra are shown in Supplementary

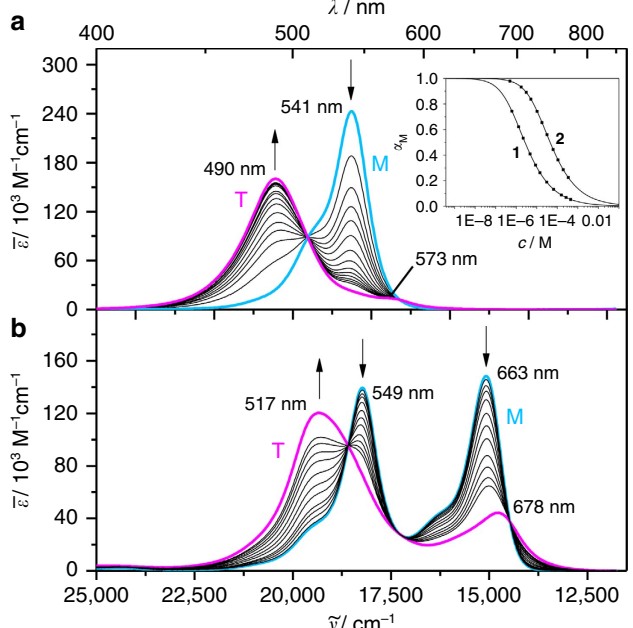

**Figure 2 | UV/vis spectroscopic studies.** Concentration-dependent UV/vis spectra of (**a**) bis(merocyanine) **1** ($c = 4.4 \times 10^{-7}$–$6.0 \times 10^{-4}$ M) and (**b**) bis(merocyanine) **2** ($c = 5.0 \times 10^{-7}$–$3.4 \times 10^{-4}$ M) in chlorobenzene at 298 K. The arrows indicate the spectral changes upon increasing the concentration. The turquoise (denoted with M) and the magenta lines (denoted with T) represent the calculated spectra of the monomers and the tetrachromophoric (dimer) stacks, respectively, by extrapolation of the spectral data towards most diluted and most concentrated solutions. Inset in **a**: Plot of fractions of monomeric species $\alpha_M$ against concentration and non-linear regression analysis of the data based on the dimerization model for bis(merocyanine) dyes **1** and **2** in chlorobenzene.

Figs 8–15. The $^1$H NMR spectrum of **1** in high-polarity solvent CD$_2$Cl$_2$ ($c = 1 \times 10^{-3}$ M), where this dye does not aggregate, displays a single set of nicely resolved proton signals as expected for the monomer of this $C_{2v}$ symmetric molecule, whose both halves are chemically equivalent on the NMR time scale (Fig. 3c). The signals could be clearly assigned to the chromophore and backbone protons of this dye by 2D correlation spectroscopy (COSY; data are listed in Supplementary Table 1). In contrast to the spectrum in CD$_2$Cl$_2$, $^1$H NMR spectrum of **1** in less-polar C$_6$D$_5$Cl ($c = 1 \times 10^{-3}$ M), where almost only aggregated species are present (inset of Fig. 2a), shows a complex signal pattern with significantly higher number of proton resonances (Fig. 3d). As the proton signals are well separated, they can be reliably assigned to the chromophore and spacer protons of dye **1** by 2D COSY and rotating frame Overhauser effect spectroscopy (ROESY) experiments, revealing a splitting of the monomer proton signals into two sets of signals (for the detailed assignment see Supplementary Table 1), with the exception for the protons 5 of two methylene groups in backbone which, instead of one singlet in monomer spectrum, show four doublets (protons 5α/β and 5′α/β) with a coupling constant of about 16 Hz each that is typical for a geminal coupling[36]. These spectral features are in compliance with a centrosymmetric dimer structure as shown in Fig. 3b. In this dimer structure with antiparallel stacking of the chromophores the two halves of each monomer are no more chemically equivalent, and thus splitting of the monomer proton signals is observed. As the methylene protons of the spacer unit are no more chemically equivalent in this dimer structure, four sets of

doublets with geminal coupling are observed. The 2D NMR spectrum obtained by ROESY measurements of dye **1** aggregate in C$_6$D$_5$Cl ($c = 1 \times 10^{-3}$ M; Supplementary Fig. 10) shows cross-peaks between the resonances of chromophore proton 3′ of one monomer and backbone proton 6 of the other monomer (as indicated with double arrow in Fig. 3b) revealing inter-molecular close spatial proximity between these protons. Thus, the results of ROESY experiments are highly supportive for a dimer structure of the aggregate.

For the unsymmetrical bis(merocyanine) **2**, in principle three different types of dimer aggregates are possible (Fig. 1b). To clarify which of these aggregates are formed, we have performed NMR studies of **2** in a 1:1 mixture of 1,1,2,2-tetrachloroethane-d$_2$/tetrachloromethane ($c = 8 \times 10^{-3}$ M) where almost only aggregated species are present as confirmed by UV/vis absorption spectroscopy (Supplementary Fig. 16). As in the case of symmetric bis(merocyanine) **1**, sharp and well resolved signals are observed in $^1$H NMR spectrum of bis(merocyanine) **2** in above-mentioned solvent mixture, indicating the formation of well-defined aggregates with definite size (Fig. 4d). Relevant proton signals could be assigned with the help of COSY and heteronuclear single-quantum coherence (HSQC) experiments (for the detailed assignment see Supplementary Table 2). No splitting of the proton signals occurs upon aggregation, while some of the signals of significant protons are up- or down-field shifted compared with those in monomer spectrum in polar CD$_2$Cl$_2$ (Fig. 4c). Based on these observations, the presence of aggregate type II dimer, which should exhibit two sets of proton signals due to the lack of symmetry, could be excluded. On the basis of $^1$H NMR spectrum type I and III dimers cannot be differentiated because both of them should show only one set of signals due to the presence of an inversion center in the dimer structure. Therefore, ROESY NMR studies were performed. In the corresponding ROESY spectrum of aggregated **2** (Supplementary Fig. 14) a cross-peak between protons 6 and 8 of the naphthalene spacer and 15 of the long merocyanine chromophore can be observed, which indicates an intercalation of the long chromophore in the stack. Moreover, no cross-peaks are observed between protons 6 and 8 of the spacer unit and protons 2, 3 and 4 of the short chromophore (Supplementary Fig. 14). Therefore, an intercalation of the short chromophore (as in type III dimer) can be ruled out and hence the formation of type I dimer with the structure shown in Fig. 4b where the long chromophores are located at the inner part of the tetrachromophoric stack is most reasonable.

**Structural elucidation by X-ray analysis.** Single crystals of bis(merocyanine) **1** suitable for X-ray analysis could be grown by slow diffusion of methanol into a chloroform solution of **1** at room temperature, while crystals of bis(merocyanine) **2** were obtained from a saturated solution in 1,4-dioxane at room temperature. As shown in Figs 3e and 4e (solvent molecules are omitted for clarity), the dimer structures we proposed for **1** and **2** based on our spectroscopic studies in solution can be confirmed in the solid state by single-crystal X-ray analysis.

In the solid state, bis(merocyanine) **1** bears a triclinic space group $P\bar{1}$ with two dye molecules among six molecules of chloroform and two molecules of methanol in one unit cell (Fig. 3e and Supplementary Fig. 17). The packing motif is described by dimeric units of the tweezer-like dyes that are arranged in one-dimensional columnar stacks, surrounded by a hydrogen-bonded network of solvent molecules. A closer look on the dimer units reveals a centrosymmetric stack comprising four antiparallel oriented chromophore units (Fig. 3e). The crystal structure of unsymmetric bis(merocyanine) **2** also shows a

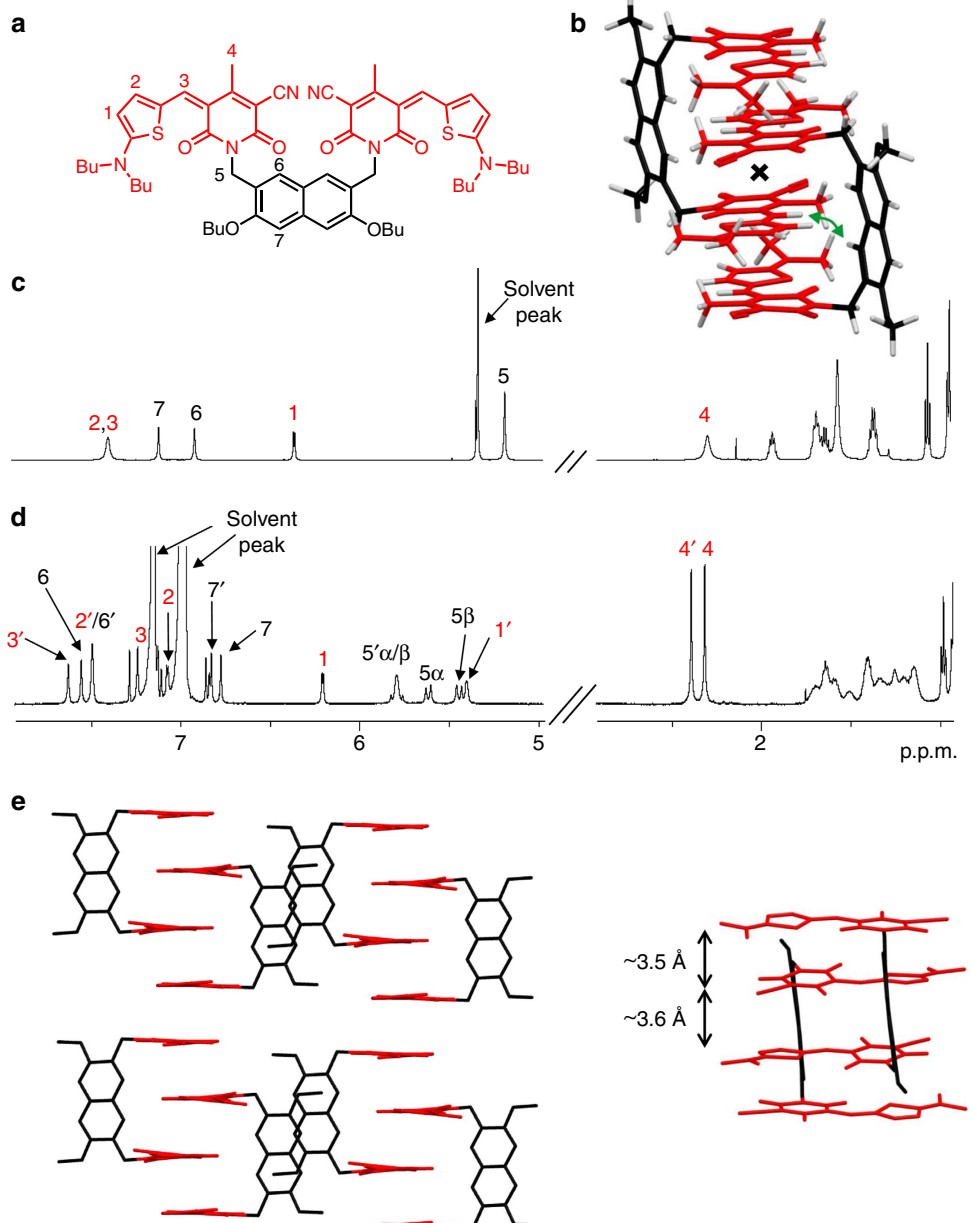

**Figure 3 | NMR studies and single-crystal X-ray analysis of bis(merocyanine) 1.** (**a**) Chemical structure of **1** with the significant protons numbered. (**b**) Geometry-optimized structure (B97D3/def2-SVP, butyl chains were replaced by methyl groups) of dimer aggregate. The inversion center of the dimer is indicated with a cross, and the green curved arrow indicates the close proximity of chromophore and backbone protons in the dimer. (**c**) Relevant sections of $^1$H NMR (600 MHz) spectrum of **1** monomer in $CD_2Cl_2$ ($c = 1 \times 10^{-3}$ M) at 295 K. (**d**) Spectrum of aggregate in $C_6D_5Cl$ ($c = 1 \times 10^{-3}$ M) at 233 K with the assignment of significant proton signals. The two sets of signals for the aggregate chromophore protons are denoted with and without prime of the respective signals. (**e**) Molecular packing of bis(merocyanine) **1** in the solid state (side view) with enlargement of the dimer structure motif (front view). Solvent molecules and hydrogen atoms are omitted for clarity and butyl chains were replaced by methyl groups.

triclinic space group $P\bar{1}$ with two dye molecules and two molecules of dioxane in the unit cell (Fig. 4e and Supplementary Fig. 18). As our NMR studies suggested, the long chromophores of two dye molecules are located in the interior and the short chromophores in the exterior of the quadruple dye stack. In contrast to bis(merocyanine) **1**, the packing motif of **2** displays no extended π-stack in the crystal since the next-neighboured bimolecular stack of four dyes is laterally shifted by 5.2 Å and transversally shifted by 3.5 Å (Fig. 4e).

The distances between the chromophores within one dimer of 3.5 and 3.6 Å for bis(merocyanine) **1** and 3.4 Å for bis(mero-cyanine) **2** confirm a tight packing at van der Waals distance,

which explains the strong aggregation tendency of both bis(merocyanine) dyes in less-polar solvents. Supported by our in-depth NMR and UV/vis spectroscopic studies, we can conclude that similar stacks as observed in the solid state are present in solution.

**Quantum chemical analysis.** Geometry optimizations of dimer structures of **1** and **2** were performed on the DFT level using the B97D3 (ref. 37) functional including dispersion correction and def2-SVP[38] as basis set. For both dimer structures an overall ground state dipole moment of zero Debye is obtained, revealing a centrosymmetric geometry with an antiparallel alignment of the

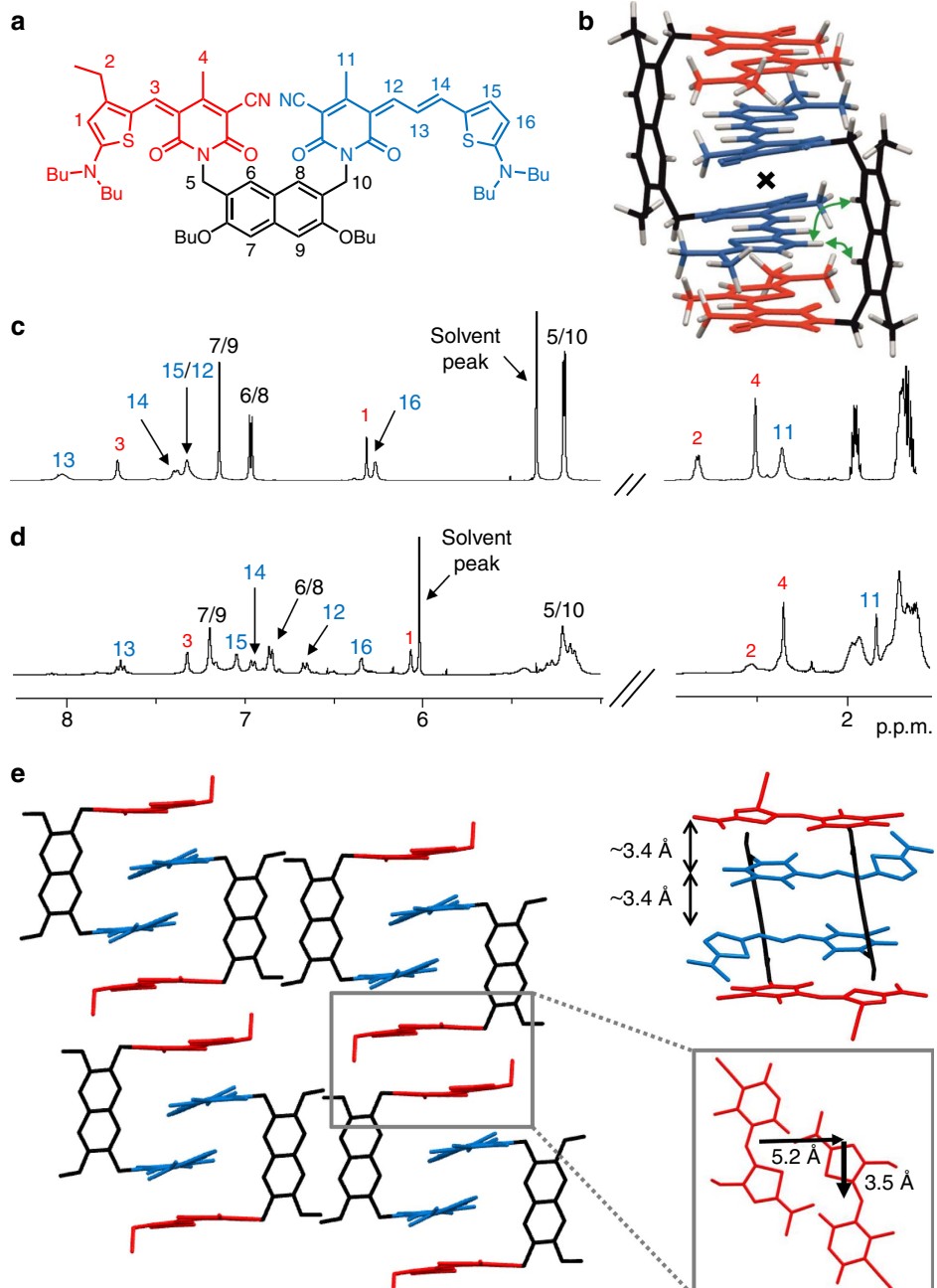

**Figure 4 | NMR studies and single-crystal X-ray analysis of bis(merocyanine) 2.** (**a**) Chemical structure of **2** with the significant protons numbered. (**b**) Geometry-optimized structure (B97D3/def2-SVP, alkyl chains were replaced by methyl groups) of dimer aggregate. The inversion center of the dimer is indicated with a cross, and the green curved arrows indicate the close proximity of chromophore and backbone protons in the dimer. (**c**) Relevant sections of $^1$H NMR (600 MHz) spectrum of **2** monomer in $CD_2Cl_2$ ($c = 5 \times 10^{-3}$ M) at 295 K. (**d**) Spectrum of aggregate in a 1:1 mixture of 1,1,2,2-tetrachloroethane-$d_2$/tetrachloromethane ($c = 8 \times 10^{-3}$ M) at 253 K with the assignment of significant proton signals. (**e**) Molecular packing of **2** in the solid state (side view) with enlargement of the dimer structure motif (front view). Solvent molecules and hydrogen atoms are omitted for clarity and butyl chains were replaced by methyl groups. A section of the crystal structure (top view) of **2** is shown (**e**, right) illustrating the lateral and transversal shift of two chromophores of next-neighboured quadruple dye stacks.

ground state dipole moments of the chromophores (Fig. 5a). On the basis of the electrostatic potential surfaces one can clearly see that the strong dipolar character of the individual merocyanine chromophores (Supplementary Fig. 19b) is much less pronounced within the aggregate stack (Fig. 5a) as it is expected for the antiparallel alignment of the dipolar chromophores. Furthermore, the highest occupied molecular orbital (HOMO) of bis(merocyanine) **1** dimer has large coefficients on all four

merocyanine chromophores (Fig. 5b, left), while the HOMO of aggregated **2** is mostly localized in the inner part of the dye stack, that is, on the long chromophores (Fig. 5b, right). This is in accordance with the observation that the long chromophore shows a lower oxidation potential than the short one and thus the former is easier to oxidize[34]. Hence, due to the same reason, the HOMO energy level of the heteroaggregate (dimer of **2**) is higher compared with that of the homoaggregate (dimer of **1**).

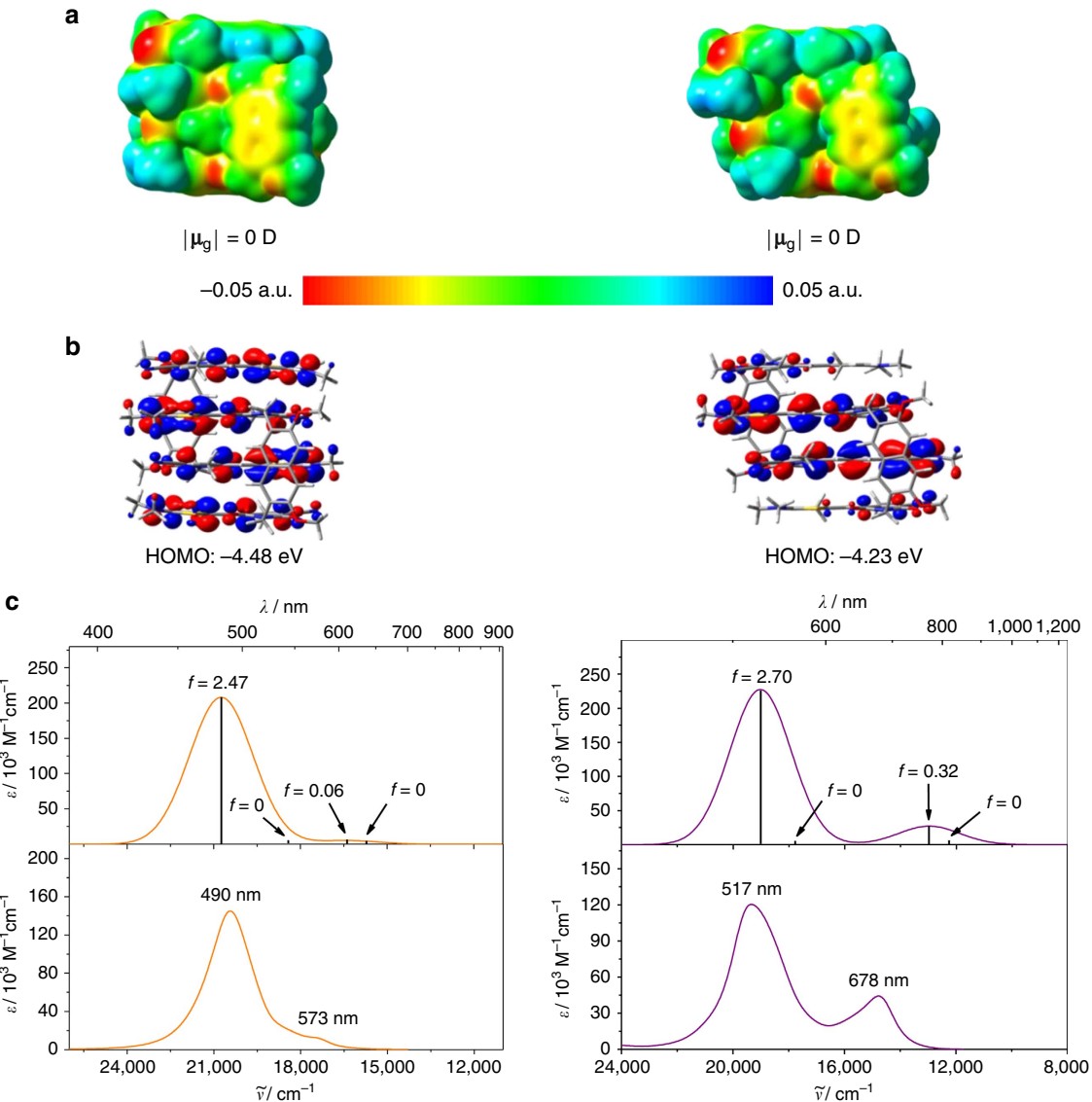

**Figure 5 | Quantum chemical calculation studies. (a)** Electrostatic potential surface (isovalue = 0.001 a.u.) and **(b)** HOMO distribution (isovalue = 0.02 a.u.) of the geometry-optimized structures (B97D3, def2-SVP) of the dimers of bis(merocyanine) dye **1** (left) and **2** (right). **(c)** time-dependent DFT (ωB97/def2-SVP) calculated (top) and experimental dimer spectra (bottom) of bis(merocyanine) **1** (left) and **2** (right). The calculated spectra were shifted by 0.6 eV towards lower energies. The experimental dimer spectra were calculated from the data obtained by concentration-dependent UV/vis studies in chlorobenzene.

To get deeper insight into the absorption properties of the quadruple merocyanine dye stacks, TDDFT calculations were performed on the geometry-optimized structures employing the long-range corrected ωB97 (ref. 39) functional (the results are listed in Supplementary Table 3). The simulated absorption spectra, together with the corresponding experimental spectra, are shown in Fig. 5c. Obviously, the shape of the experimental dimer spectra derived from UV/vis experiments are well reproduced by the TDDFT calculations and one can easily recognize that for both aggregate structures the transition to the lowest and third excited state is forbidden (oscillator strength $f$ is equal to zero for both states). On the other hand, the highest excited state transition is strongly allowed for both quadruple dye stacks as reflected by a very intense absorption band. The experimental dimer spectrum of the homoaggregate exhibits an additional weak absorption shoulder at longer wavelength (573 nm), which can be attributed to the partially allowed transition to the second excited state (Fig. 5c, left). This transition

is significantly more allowed in the heteroaggregate dye stack that shows a stronger absorption at the longer wavelength of 678 nm (Fig. 5c, right).

For more than 50 years absorption properties of dye aggregates consisting of identical chromophores (homoaggregates) are commonly analysed based on Kasha's theory by assuming an exciton coupling between the transition dipole moments of the constituent chromophore units[1,9,40,41]. The exciton theory can be classified as a perturbation theory, in which the chromophores maintain their individuality and the exciton coupling is described as an interaction between the oscillating transition dipole moments of the localized transitions. Such coupling leads to a splitting of the excited state energies, which results in different absorption features of the aggregates compared to the respective monomeric chromophores. Here we apply Kasha's exciton theory to heteroaggregates, and for comparison also to homoaggregates, consisting of four chromophore units, respectively, to analyse the absorption properties of the dimer structures of bis(merocyanine)

dyes **1** and **2**. In particular, we want to elucidate whether the spectral changes of **2** upon aggregation are caused by exciton coupling not only between the same types of chromophores but also between the chromophores of different types, that is, chromophores with different excitation energies. The detailed description of our quantum chemical analyses can be found in Supplementary Discussion, Supplementary Tables 4–9 and Supplementary Notes 1 and 2. The resulting exciton state diagrams for the homo- and heteroaggregate dye stacks are displayed in Fig. 6.

Four non-degenerated excited states can be observed as predicted by TDDFT calculations. The amount of nodes between the oscillating transition dipole moments of the chromophores is decreasing from three for the lowest excited state to zero for the highest excited state. For both aggregate structures, the transition to the highest excited state is strongly allowed since all the

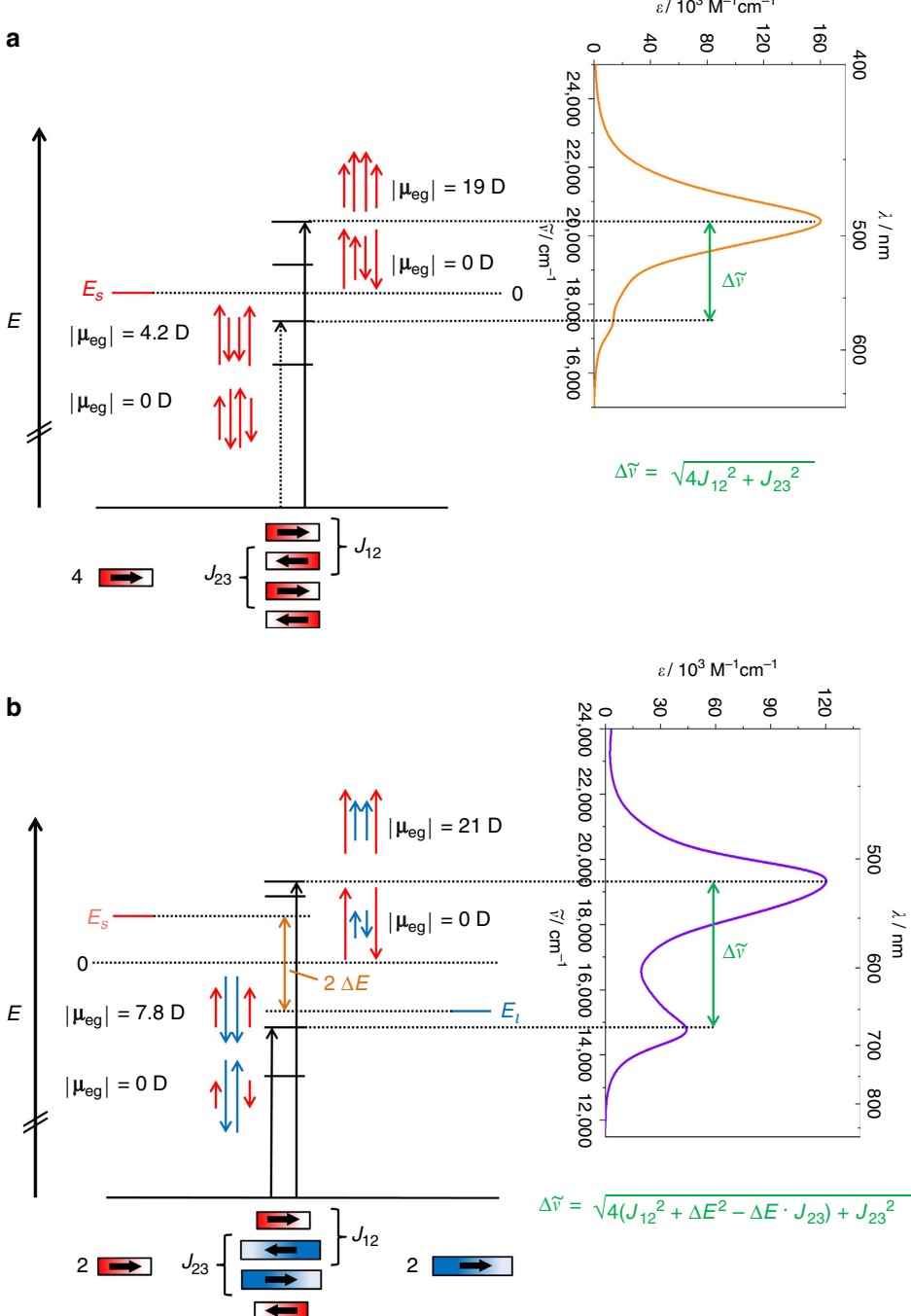

**Figure 6 | Exciton state diagram and UV/vis absorption spectra.** State diagram and spectra of the quadruple dye stacks of (**a**) bis(merocyanine) **1** and (**b**) bis(merocyanine) **2**. The arrows (red for the short chromophore and blue for the long chromophore, respectively) indicate the phase relations between the corresponding transition dipole moments. The lengths of the arrows reflect the magnitude of the coefficients of the oscillating transition dipole moments. The corresponding experimental dimer spectrum obtained from concentration-dependent UV/vis spectroscopy in chlorobenzene is depicted to indicate the allowed transitions. $J_{12}$ and $J_{23}$ denote the exciton coupling energies for the chromophore pairs in the exterior and in the interior of the dye stack, respectively, while $E_s$ and $E_l$ are the excited state energies of the short and long chromophore, respectively.

transition dipole moments of the chromophores are oscillating in phase resulting in a large magnitude of the overall transition dipole moment ($|\mu_{eg}|$). Furthermore, the excitation to the lowest and third excited state is forbidden since the overall transition dipole moment amounts to zero. This is due to the fact that the transition dipole moments that oscillate with the same coefficients (note: the magnitude of the coefficient is indicated by the length of the arrows that represent the transition dipole moments) are out of phase. On the other hand, the transition to the second lowest excited state is partially allowed for both aggregate structures since the transition dipole moments of the outer chromophores are oscillating with different coefficients than the transition dipole moments of the inner chromophores resulting in overall transition dipole moments of 4.2 D (for **1**) and 7.8 D (for **2**). This value is higher for the heterodimer of bis(merocyanine) **2** because the long chromophore exhibits a larger transition dipole moment of 12.3 D than the short chromophore having a value of 9.9 D (Supplementary Fig. 19c). Thus, the dimer spectrum of bis(merocyanine) **1** shows only a weak absorption shoulder at higher wavelength (573 nm), while the corresponding absorption band (678 nm) is more intense for bis(merocyanine) **2** dimer. The energy splittings $\Delta\tilde{v}$ between the two absorption bands of the quadruple dye stacks depend on the exciton coupling energies $J$ between next-neighbouring chromophores within the dye stacks as depicted in Fig. 6. The exciton coupling energies were calculated by applying the transition charge method[42] (Supplementary Discussion and Supplementary Tables 4 and 5) to give $J_{12} = 1{,}576$ cm$^{-1}$ and $J_{23} = 1{,}452$ cm$^{-1}$ for the homoaggregate, where $J_{12}$ and $J_{23}$ denote the coupling energies for the chromophore pair in the exterior and in the interior of the stack, respectively (Supplementary Table 6). In the case of the heteroaggregate the exciton coupling energies amount to $J_{12} = 1{,}861$ cm$^{-1}$ and $J_{23} = 1{,}108$ cm$^{-1}$, respectively (Supplementary Table 6). Accordingly, the largest coupling is indeed given for the heteropair despite of the significantly different excitation energies of the given chromophores. The positions of the calculated absorption maxima (477 and 572 nm for **1** and 517 and 663 nm for **2**, respectively (Supplementary Table 7)) are in good agreement with the ones of the experimental dimer spectra obtained by concentration-dependent UV/vis measurements in chlorobenzene. Furthermore, the oscillating transition dipole moments shown in Fig. 6 correctly describe the transition densities of the excited states obtained by TDDFT calculations (Supplementary Fig. 20).

## Discussion

The UV/vis absorption properties of the quadruple $\pi$-stack homoaggregate of bis(merocyanine) **1** can be rationalized by Kasha's exciton theory assuming an exciton coupling between the four identical chromophores. Hence, the quadruple $\pi$-stack can be classified as H-type aggregate exhibiting a hypsochromic shift of the absorption with respect to the monomeric chromophore. In contrast, the absorption spectrum of the quadruple $\pi$-stack heteroaggregate of bis(merocyanine) **2** is more complex. It shows a hypsochromically shifted main absorption band of high intensity and a second band of considerably strong intensity and a bathochromic shift. This observation cannot be rationalized by presuming an exciton coupling only between the same types of chromophores which should result in two hypsochromically shifted absorption bands with respect to the absorption bands of the corresponding monomeric chromophores. Our extension of the molecular exciton theory to heteroaggregates adequately describes the absorption spectrum of the aggregate of bis(merocyanine) **2** and is in accordance with the results obtained by TDDFT calculations. Thus, exciton coupling in

$\pi$-stacked dyes is not only present between the same type of chromophores, as is often assumed, and significantly influences the optical absorption properties of the heteroaggregate. Our extension of Kasha's theory for hetero-$\pi$-stacks may be applied to larger homo- and hetero-aggregates and enable the design of desirable absorption features and to predict materials' properties for bulk systems of larger complexity as required in many research fields including (opto-)electronics and photovoltaics. Thus, we are currently focusing on the construction of well-defined supramolecular $\pi$-stacks of larger size based on dipolar interactions.

## Methods

**General.** All solvents and reagents were purchased from commercial sources and used as received without further purification. Microwave reactions were performed in Discover CEM Microwave Reactor. Column chromatography was performed using silica gel 60 M (0.04–0.063 mm). NMR spectra were recorded on an Advance 400 or Advance DMX 600 spectrometer at 295 K, unless otherwise stated. The spectra were calibrated to the residual solvent peak and the chemical shifts $\delta$ are given in p.p.m. Multiplicities are denoted as follows: s, singlet, d, doublet, t, triplet, q, quartet, dd, doublet of doublets, m, multiplet, br, broad. High-resolution mass spectra (ESI) were recorded on an ESI MicrOTOF Focus spectrometer. Elemental analysis were performed on a CHNS 932 analyzer.

**UV/vis spectroscopy.** For all spectroscopic measurements spectroscopic grade solvents (Uvasol) were used. Concentration-dependent UV/vis spectra were recorded on UV/vis spectrometers Lambda 950 or 40P. The spectral band width and the scan rate were 1 nm and 120 nm min$^{-1}$, respectively. Stock solutions of the studied compounds in chlorobenzene were accurately prepared, and continuously diluted for absorption measurements at different concentrations by taking into account the solubility and the absorbance of the respective compound. The UV/vis measurements were performed in conventional quartz cell cuvettes with path lengths of 0.2–50 mm.

**Computational details.** Computational calculations were performed using the Gaussian 09 program package[43]. Geometry optimizations were carried out at the DFT level for the dimer aggregates of bis(merocyanine) **1** and **2** (butyl and ethyl chains were replaced by methyl groups) as well as for reference compounds **Ref 1** and **Ref 2** (Supplementary Fig. 19) with B97D3 (ref. 37) as functional and def2-SVP[38] as basis set. The structures were geometry optimized, followed by frequency calculations on the optimized structures. One very small imaginary frequency of 13i cm$^{-1}$ was obtained for the dimers of each of the bis(merocyanine) dyes **1** and **2**. Small imaginary frequencies ($<100$i cm$^{-1}$) are considered most likely to be an artefact of the calculation[44], thus the resulting geometries can be seen as real minima.

TDDFT calculations were performed on the geometry-optimized structures using the long-range corrected $\omega$B97 functional[39] and def2-SVP[38] as basis set.

UV/vis spectra of the dimer aggregates of bis(merocyanine) dyes **1** and **2** were simulated with the help of the GaussView 5 (ref. 45) visualization software package using the results obtained by TDDFT calculations with a half-width at half-height of 0.16 eV.

**Single-crystal X-ray analysis.** Single-crystal X-ray diffraction data for bis(merocyanine) **1** were collected at 100 K on a Bruker X8APEX-II diffractometer with a CCD area detector and multi-layer mirror monochromated MoK$_\alpha$ radiation. The data for bis(merocyanine) **2** were collected at 100 K on a Bruker D8 Quest Kappa Diffractometer with a Photon100 CMOS detector and multi-layered mirror monochromated CuK$_\alpha$ radiation. The structures were solved by direct methods, expanded with Fourier techniques and refined with the Shelx software package[46]. All non-hydrogen atoms were refined anisotropically. Hydrogen atoms were included in the structure factor calculation on geometrically idealized positions.

*Crystal data* for bis(merocyanine) **1** ($C_{64}H_{81}Cl_9N_6O_7S_2$): $M_r = 1{,}429.51$, purple-blue block, $0.50 \times 0.50 \times 0.40$ mm$^3$, triclinic space group P$\bar{1}$, $a = 13.437(2)$ Å, $\alpha = 74.010(5)°$, $b = 14.1187(19)$ Å, $\beta = 71.294(3)°$, $c = 19.899(2)$ Å, $\gamma = 85.409(2)°$, $V = 3{,}437.3(8)$ Å$^3$, $Z = 2$, $\rho(calcd.) = 1.381$ g cm$^{-3}$, $\mu = 0.483$ mm$^{-1}$, $F(000) = 1{,}496$, $GooF(F^2) = 1.040$, $R_1 = 0.0591$, $wR^2 = 0.1649$ for $I > 2\sigma(I)$, $R_1 = 0.0746$, $wR^2 = 0.1797$ for all data, 14,143 unique reflections [$\theta \le 25.242°$] with a completeness of 100.0% and 894 parameters, 209 restraints.

Crystal data for bis(merocyanine) **2** ($C_{64}H_{80}N_6O_6S_2 \cdot C_4H_8O_2$): $M_r = 1{,}181.56$, blue block, $0.32 \times 0.16 \times 0.12$ mm$^3$, triclinic space group P$\bar{1}$, $a = 13.9822(3)$ Å, $\alpha = 74.6400(11)°$, $b = 15.9375(4)$ Å, $\beta = 65.1290(10)°$, $c = 16.5230(4)$ Å, $\gamma = 69.0340(10)°$, $V = 3{,}090.79(13)$ Å$^3$, $Z = 2$, $\rho(calcd.) = 1.270$ g cm$^{-3}$, $\mu = 1.267$ mm$^{-1}$, $F(000) = 1{,}268$, $GooF(F^2) = 1.087$, $R_1 = 0.0584$, $wR^2 = 0.1515$ for $I > 2\sigma(I)$, $R_1 = 0.0693$, $wR^2 = 0.1583$ for all the data, 12,419 unique reflections ($\theta \le 74.643°$) with a completeness of 98.9% and 800 parameters, 12 restraints.

**Data availability.** Crystallographic data have been deposited with the Cambridge Crystallographic Data Centre as supplementary publication no. CCDC 1476632 (**1**); CCDC 1476633 (**2**). These data can be obtained free of charge from The Cambridge Crystallographic Data Centre via www.ccdc.ac.uk/data.request/cif. The authors declare that all other data supporting the findings of this study are available within the article and its Supplementary Information files.

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

## Acknowledgements

D. B. thanks the Fonds der Chemischen Industrie for a PhD scholarship. We thank Dr C. Burschka for his help regarding the X-ray analysis of bis(merocyanine) **1**.

## Author contributions

F.W. conceived the project and suggested the suitable molecules; D.B. and A.Z.-K. synthesized the molecules and performed most experiments; E.K. and A.Z.-K. carried out NMR experiments; D.S. performed the X-ray analysis; D.B. performed all theoretical studies; D.B., E.K. and F.W. co-wrote the paper.

## Additional information

**Competing financial interests:** The authors declare no competing financial interests.

