## [Peer review file · Nature Communications]

Reviewers' comments:

Reviewer #1 (Remarks to the Author):

This is an interesting approach to a rather mature research topic that leads to the generation of a novel type of heteroaggregate structure. The two key dimers studied in this work have been very well characterised - this is the main attraction of the work - thereby facilitating detailed experimental and theoretical investigations. The work has been done carefully and thoroughly - perhaps a bit too much detail is provided and the text becomes rather repetitious in places. The various calculations seem to agree remarkably well with experiment, especially the ZINDO(S) results which are often out-of-line with experiment. The conclusions are fine but not very informative. The strength of the work relates to the nice structures and their elucidation. It's difficult to say if the work has given a deeper understanding of exciton coupling - it rather looks as if the existing theory does a good job.

Overall, this is a very nice contribution to a topical subject. The text is too long, especially the introduction, and would benefit by some serious pruning. The figures are excellent. Some clarification about how well existing Jahn-Teller theory works for 4-chromophore dimers and for heterodimers would help the nonspecialist reader.

Reviewer #2 (Remarks to the Author):

The paper I had to review deals with homo- and heteroaggregate quadruple stacks of merocyanine dyes. The novelty of the work lies in the direct comparison of heteroaggregate stacks formed from unsymmetric bis-merocyanines and homoaggregates of symmetric bis-merocyanines. The experimental results from UV-Vis spectroscopy are compared to and explained by quantum chemistry calculations regarding the excitonic coupling in the different stacks. There, the authors claim that their heteroaggregates are the first thoroughly characterized multichromophoric dye stacks showing strong exciton coupling.

I agree with the authors that selfassembly of different chromophores into heteroaggregates can lead to systems with exceptional properties that cannot be achieved by using chromophores of the same type. However, as the authors come from the field of solvent chemistry, they are probably not aware of the fact that since some time highly ordered (mono-)layers of two different organic dyes are being prepared by organic molecular beam epitaxy / deposition, resulting in epitaxial stacks of different chromophores, and are investigated by surface sensitive optical spectroscopies to investigate in detail the excitonic interaction between the two different chromophores in the layer stack. That relativizes the "synthetic challenge" mentioned in Line 39. Since even longer time, the excitonic coupling in epitaxial stacks of one chromophore has been tackled by both experimentalists and theorists. There are numerous examples in literature, and some are summarized in a review by Forker et al. (Annu. Rep. Prog. Chem., Sect. C: Phys. Chem. 108, 34-68 (2012)).

Having said so, I still admire the work presented in the manuscript for sake of clarity, and I feel that the results can and will also stimulate the work on highly ordered films, not only on solution grown aggregates.

The concentration dependent UV-Vis spectroscopy is obviously done with great care (as revealed by the clear appearance of isosbestic points), and analyzed thoroughly. The excitonic coupling between the merocyanines of different chromophore length is clearly evidenced.

While the quantum chemistry calculations are on the well-known ZINDO/S level and are more or less standard, the detailed analysis based on the Kasha model and resulting in useful values for the exciton coupling energy J is valuable for the reader.

To summarize, my appraisal of the manuscript is a little bit ambivalent. On one side, all methods (theoretical and experimental) and the general topic "dye aggregates" are well-known to the community, therefore appearing to be not too fancy. On the other side, the success of having synthesized and thoroughly analyzed a multichromophoric heteroaggregate is indeed new and worth publishing. The Würthner group is known to have high expertise in the field of dye aggregates, and consequently the entire manuscript is sound and solid. Therefore, I recommend publication in Nat. Comm. The publication criteria "Nature Communications is committed to publishing important advances of significance to specialists within each field." is clearly fulfilled here.

Finally I have to mention that I did not check the crystallographic data for errors / problems because that does not fall into my area of expertise, nor can I comment on the sections regarding the synthesis of the materials and NMR.

Torsten Fritz

Reviewer #3 (Remarks to the Author):

The paper by Bialas et al. presents a combined experimental and theoretical study of the excitonic properties of homo and hetero aggregates of merocyanines.

The experimental part of the study is interesting and well presented while the quantum-mechanical analysis is confused and, in its present form, it does not add anything to the paper. Therefore, I strongly suggest the authors to largely revised it according to the following points, before the paper can be considered for publication.

1) The authors should check if the ZINDO approach is suited for this kind of systems by validating it against a more accurate method. As a matter of fact, I do not understand why the authors use a DFT approach for the geometry optimization and then they limit the excited state analysis of the dimers to ZINDO.

2) The authors should compare the results for the aggregate with those of the monomers to understand if the level of calculation used correctly reproduces the aggregation effects. In other words, are the energy splittings calculated by ZINDO accurate enough? Have the authors tried to compare the resulting couplings (J) with a dipole-dipole approximation or with other approaches based, for example, on transition charges obtained with more accurate QM methods?

3) The authors should clarify better why they need an excitonic approach when they can directly compute the properties of the full system. In any case, the description of the excitonic approach should be moved to the SI, leaving in the main text only the numerical results summarized in Figure 6.

Finally, even if I understand that for these systems the transition dipole is along the main axis, I strongly suggest to rewrite eq.(5) in its vectorial form.

Responses to the referees' comments

Reviewer 1:

Comments: This is an interesting approach to a rather mature research topic that leads to the generation of a novel type of heteroaggregate structure. The two key dimers studied in this work have been very well characterised - this is the main attraction of the work - thereby facilitating detailed experimental and theoretical investigations. The work has been done carefully and thoroughly - perhaps a bit too much detail is provided and the text becomes rather repetitious in places. The various calculations seem to agree remarkably well with experiment, especially the ZINDO(S) results which are often out-of-line with experiment. The conclusions are fine but not very informative. The strength of the work relates to the nice structures and their elucidation. It's difficult to say if the work has given a deeper understanding of exciton coupling - it rather looks as if the existing theory does a good job.

Overall, this is a very nice contribution to a topical subject. The text is too long, especially the introduction, and would benefit by some serious pruning. The figures are excellent. Some clarification about how well existing Kasha-type theory works for 4-chromophore dimers and for heterodimers would help the nonspecialist reader.

Answer: Following the reviewer's suggestions we have shortened the manuscript by pruning the introduction. Thus, the discussion of the natural light harvesting complexes has been completely removed and some other parts condensed in the newly written paragraph "Notably, it is often ..." (marked in yellow). Furthermore, the theoretical analysis based on exciton coupling is moved into the Supplementary Information. The part "Results and Discussion" was redefined as "Results" while the former part "Conclusions" was renamed to "Discussion" and revised.

Concerning existing Kasha-type theories: The novelty of our manuscript is to extend Kasha's exciton theory to heteroaggregate systems by confirming an exciton coupling also between different types of chromophores with different excitation energies. It is commonly believed that only identical chromophores (i.e. of the same excitation energy) exhibit strong exciton coupling. However, this does not comply with our results obtained for the heteroaggregate of bis(merocyanine) **2**. We are convinced that this unprecedented result is now better highlighted in our revised discussion. In particular, we have further elaborated Figure 6 and added a paragraph on the J_{12} and J_{23} coupling energies for homo and hetero dye pairs. Whilst the field of homo-aggregates is mature, properties of hetero-aggregates are indeed not well investigated. One reason is for sure that they are very difficult to obtain and to characterize unambiguously. This is the reason why we spent much effort on this and have included all structural evidence in our manuscript. We thank all reviewers for highlighting this point in their reports.

Reviewer 2:

Comments: The paper I had to review deals with homo- and heteroaggregate quadruple stacks of merocyanine dyes. The novelty of the work lies in the direct comparison of heteroaggregate stacks formed from unsymmetric bis-merocyanines and homoaggregates of symmetric bis-merocyanines. The experimental results from UV-Vis spectroscopy are compared to and explained by quantum chemistry calculations regarding the excitonic coupling in the different stacks. There, the authors claim that their heteroaggregates are the first thoroughly characterized multichromophoric dye stacks showing strong exciton coupling.

I agree with the authors that selfassembly of different chromophores into heteroaggregates can lead to systems with exceptional properties that cannot be achieved by using chromophores of the same type. However, as the authors come from the field of solvent

chemistry, they are probably not aware of the fact that since some time highly ordered (mono-)layers of two different organic dyes are being prepared by organic molecular beam epitaxy / deposition, resulting in epitaxial stacks of different chromophores, and are investigated by surface sensitive optical spectroscopies to investigate in detail the excitonic interaction between the two different chromophores in the layer stack. That relativizes the "synthetic challenge" mentioned in Line 39. Since even longer time, the excitonic coupling in epitaxial stacks of one chromophore has been tackled by both experimentalists and theorists. There are numerous examples in literature, and some are summarized in a review by Forker et al. (Annu. Rep. Prog. Chem., Sect. C: Phys. Chem. 108, 34-68 (2012).).

Answer: We were indeed not aware of this rather different "physical" approach to obtain heteroaggregates. Thus, as suggested we removed the comment on the synthetic challenges regarding the preparation of heteroaggregates. Furthermore, we included the review article of Forker et al. as a reference in the introduction when elucidating strategies for the design of well-defined π -stacks. Furthermore, two additional references of Broch et al. and Reinhardt et al. were added that are concerned with intermolecular coupling between different types of chromophores in mixed films.

Having said so, I still admire the work presented in the manuscript for sake of clarity, and I feel that the results can and will also stimulate the work on highly ordered films, not only on solution grown aggregates.

The concentration dependent UV-Vis spectroscopy is obviously done with great care (as revealed by the clear appearance of isosbestic points), and analyzed thoroughly. The excitonic coupling between the merocyanines of different chromophore length is clearly evidenced.

While the quantum chemistry calculations are on the well-known ZINDO/S level and are more or less standard, the detailed analysis based on the Kasha model and resulting in useful values for the exciton coupling energy J is valuable for the reader.

To summarize, my appraisal of the manuscript is a little bit ambivalent. On one side, all methods (theoretical and experimental) and the general topic "dye aggregates" are well-known to the community, therefore appearing to be not too fancy. On the other side, the success of having synthesized and thoroughly analyzed a multichromophoric heteroaggregate is indeed new and worth publishing. The Würthner group is known to have high expertise in the field of dye aggregates, and consequently the entire manuscript is sound and solid. Therefore, I recommend publication in Nat. Comm. The publication criteria "Nature Communications is committed to publishing important advances of significance to specialists within each field." is clearly fulfilled here.

Finally I have to mention that I did not check the crystallographic data for errors / problems because that does not fall into my area of expertise, nor can I comment on the sections regarding the synthesis of the materials and NMR.

Answer: We appreciate these comments and are grateful for the constructive criticism. It was particularly enlightening for us to get a view from a rather different perspective, i.e. preparation of heteroaggregate dye systems by molecular beam epitaxy which is an entirely different approach to the one pursued by us.

Reviewer 3:

The paper by Bialas et al. presents a combined experimental and theoretical study of the excitonic properties of homo and hetero aggregates of merocyanines.

The experimental part of the study is interesting and well-presented while the quantum-mechanical analysis is confused and, in its present form, it does not add anything to the paper. Therefore, I strongly suggest the authors to largely revise it according to the following points, before the paper can be considered for publication.

1) The authors should check if the ZINDO approach is suited for this kind of systems by validating it against a more accurate method. As a matter of fact, I do not understand why the authors use a DFT approach for the geometry optimization and then they limit the excited state analysis of the dimers to ZINDO.

Answer: As suggested, the ZINDO/S calculations were replaced by more accurate time-dependent DFT calculations. Best results were thus obtained with the long-range corrected hybrid ω B97 functional. Employing a larger basis set (def2-TZVP) and including solvent effects did not significantly change the spectra of the dimer aggregates shown in Figure 5c.

2) The authors should compare the results for the aggregate with those of the monomers to understand if the level of calculation used correctly reproduces the aggregation effects. In other words, are the energy splittings calculated by ZINDO accurate enough? Have the authors tried to compare the resulting couplings (J) with a dipole-dipole approximation or with other approaches based, for example, on transition charges obtained with more accurate QM methods?

Answer: Following the reviewer's suggestion we determined the exciton coupling energies using a more accurate method employing transition charges. For the homoaggregate, similar values were obtained with respect to the values calculated by the ZINDO/S approach while larger deviations were obtained for the heteroaggregate. Hence, the ZINDO/S calculations were replaced by the more accurate transition charge method (see Supplementary Information).

3) The authors should clarify better why they need an excitonic approach when they can directly compute the properties of the full system. In any case, the description of the excitonic approach should be moved to the SI, leaving in the main text only the numerical results summarized in Figure 6.

Answer: An additional sentence has been added on page 15 to explain our motivation for the analysis based on molecular exciton theory.

*"In particular, we want to elucidate whether the spectral changes of **2** upon aggregation are caused by exciton coupling not only between the same types of chromophores but also between the chromophores of different types."*

Furthermore, we are convinced that our revised discussion will contribute to substantiate our approach. Following the reviewers suggestion, the analysis based on molecular exciton theory was moved into the Supplementary Information.

Finally, even if I understand that for these systems the transition dipole is along the main axis, I strongly suggest to rewrite eq.(5) in its vectorial form.

Answer: We are thankful for this advice and accordingly rewrote the respective equation in the vectorial form (see Equation S9 in the Supplementary Information).

We like to express our gratitude to all three reviewers for their helpful advices. Obviously, the reviewers have very different scientific backgrounds and thus we could receive helpful advices for the improvement of different parts of our manuscript. We are glad that – despite of their different backgrounds – all reviewers showed appreciation for our “synthetic” approach and the accomplishments on our scientific pathway towards well-defined large scale dye aggregates of precisely defined supramolecular structure.

REVIEWERS' COMMENTS:

Reviewer #1 (Remarks to the Author):

The revised manuscript is an improvement and appears to meet all of the requirements necessary for publication in the journal. The authors have made an important contribution to a topical subject. This work is comprehensive and reported in a concise manner.

Reviewer #3 (Remarks to the Author):

The authors have satisfactorily addressed all the critical points present in the original version of the manuscript.

The revised version is now acceptable for publication.

Responses to the referees' comments

Manuscript NCOMMS-16-10669A: "Structural and quantum chemical analysis of exciton coupling in homo- and heteroaggregate stacks of merocyanines"

Reviewer 1:

Comments: The revised manuscript is an improvement and appears to meet all of the requirements necessary for publication in the journal. The authors have made an important contribution to a topical subject. This work is comprehensive and reported in a concise manner.

Reviewer 3:

Comments: The authors have satisfactorily addressed all the critical points present in the original version of the manuscript. The revised version is now acceptable for publication.

*Answer: We are grateful for the constructive criticism of all reviewers that has significantly improved our manuscript. Therefore, we highly appreciate that our revised manuscript has definitely been recommended for publication in *Nature Communications*.*